# Neurofilament Expression as a Biomarker of Post-COVID-19 Sudden Sensorineural Hearing Loss

**DOI:** 10.3390/diseases11030092

**Published:** 2023-06-29

**Authors:** Federica Zoccali, Carla Petrella, Maria Antonella Zingaropoli, Marco Fiore, Massimo Ralli, Antonio Minni, Christian Barbato

**Affiliations:** 1Department of Sense Organs DOS, Sapienza University of Rome, Viale del Policlinico 155, 00161 Rome, Italy; federica.zoccali@uniroma1.it (F.Z.); carla.petrella@cnr.it (C.P.); massimo.ralli@uniroma1.it (M.R.); 2Institute of Biochemistry and Cell Biology (IBBC), National Research Council (CNR), Sapienza University of Rome, Viale del Policlinico 155, 00161 Rome, Italy; marco.fiore@cnr.it; 3Department of Public Health and Infectious Diseases, Sapienza University of Rome, Viale del Policlinico 155, 00185 Rome, Italy; 4Division of Otolaryngology-Head and Neck Surgery, Ospedale San Camillo de Lellis, ASL Rieti-Sapienza University, Viale Kennedy, 02100 Rieti, Italy

**Keywords:** post-COVID-19, sudden sensorineural hearing loss (SSHL), neurofilament light chain (NfL), biomarker

## Abstract

Sudden sensorineural hearing loss (SSHL) affects a patient’s quality of life and requires rapid treatment. The etiology is viral, vascular, and autoimmune, even though, in most cases, it remains idiopathic SSHL. Since 2019, several different complications have been identified following COVID-19 infection. The post-COVID-19 ENT manifestations reported in the literature are sore throat, headache, pharyngeal erythema, nasal obstruction, rhinorrhea, upper respiratory tract infection, and tonsil enlargement. Cases of SSHL, vestibular neuronitis, and audio-vestibular disorders (such as tinnitus, dizziness, and vertigo) have also been reported, albeit in a smaller percentage of patients. We reported our experience of a case of post-COVID-19 SSHL in the absence of any other type of post-COVID symptoms or brain and internal auditory canal magnetic resonance imaging and magnetic resonance angiography modifications. We aimed to identify a serological biomarker of sudden sensorineural hearing loss, and we also dosed and monitored the value of the serum neurofilament light (NfL). the best of our knowledge, this is the first report that associates SSHL and the serological increase in NfL as a potential biomarker of neuronal-disease-related damage.

## 1. Introduction

COVID-19 is a severe acute respiratory syndrome induced by Severe Acute Respiratory Syndrome Coronavirus 2 (SARS-CoV-2), which has rapidly become one of the greatest challenges of our century, causing great medical and socioeconomic upheaval [1,2]. The pandemic was a global health concern that required a rapid biomedical response to limit the spread of the disease and deaths. An important issue in SARS-CoV-2 infection is that the virus is mutating constantly, and since the beginning of the pandemic, a plethora of variants, such as Alpha, Beta, Delta, and Omicron, have been discovered [3,4,5]. SARS-CoV-2 infection is characterized by a spectrum of clinical manifestations that frequently include dry cough, fever, and fatigue, often associated with pulmonary involvement that can progress to acute respiratory distress syndrome. Ear, nose, and throat (ENT) manifestations of COVID-19 are not common. The most common ENT symptoms reported in the literature are sore throat, smell dysfunction, headache, pharyngeal erythema, nasal obstruction, rhinorrhea, upper respiratory tract infection, and tonsil enlargement [6]. A smaller number of cases of SSHL, vestibular neuronitis, and vestibular disorders, such as vertigo, dizziness, tinnitus, have also been reported [6,7,8]. To date, different systematic reviews have addressed the theme of ENT manifestations in COVID-19 patients. To date, according to the World Health Organization, there have been 767 million confirmed cases of COVID-19, including 6.9 million deaths [9], and since 21 February 2022, a total of 13,375,580,553 vaccine doses have been administered [10]. According to the National Institute for Deafness and Communication Disorders, SSHL is defined as the rapid onset of hearing loss of at least 30 decibels across three consecutive frequencies within 72 decibels. SSHL’s etiopathogenesis is attributed to viral infections, vascular impairment, and autoimmune disorders, even though in 90% of the patients SSHL is characterized as idiopathic due to its no-cause identification, despite investigations. Wichova et al. argued that SSHL increased between 2019 and 2021, suggesting an association between COVID-19 and SSHL [11]. We report the case of a subject who developed SSHL without other types of post-COVID infection deficits after a COVID-19 infection. The patient was treated via intratympanic steroid administration without a significant restoration of their hearing loss. In this context, with the aim of exploring a potentially valuable biomarker of this neurosensorial impairment, we measured the serum levels of the neurofilament light chain protein (NfL) in the patient’s serum at the time of diagnosis before steroid administration and after therapy.

Blood-based serum NfL can be considered an easily accessible biomarker of prognosis and treatment response in patients affected by chronic neurological diseases, such as neurodegenerative disease, including Alzheimer’s, mild cognitive impairment, and tauopathies, as well as acute neurological diseases, vascular and post-traumatic injuries, and concussion. However, this is also true for patients with COVID-19-related peripheral nerve injury [12]. The neurofilament light chain is a subunit of cylindrical protein neurofilaments, a widely expressed protein family resident in large-caliber myelinated axons [12,13].

NfL is only distributed in the neuronal cytoplasm, conferring structural stability and allowing for the radial outgrowth of myelinated axons. NfL belongs to the intermediate filament protein family, which consists of three subunits based on molecular weight: neurofilament light (NfL), medium (NfM), and heavy (NfH), as well as α-internexin and peripherin [12]. Under physiological conditions, low concentrations of NfL are continuously released from axons into the cerebrospinal fluids (CSFs) and conveyed to the blood at lower concentrations across the blood–brain barrier, increasing in an age-dependent manner [14]. Because of neurodegenerating processes, inflammation, and traumatic injury, the release process of NfL is accelerated. The overflow of NfL into the interstitial fluid and contiguous inflow into the CSF and blood is the basis for its use as a biomarker [15,16]. Interestingly, there is very little evidence for an effect of blood–brain barrier impairment on the plasma or serum NfL concentration [17]. Furthermore, it is important to highlight that many psychiatric diseases not characterized by neurodegenerative processes express normal levels of NfL [18,19]. This difference between chronic neurological and psychiatric pathologies could support the differential diagnosis between cerebral dementias and alterations of the emotional and cognitive profiles [20]. There is copious evidence that blood sNfL levels reflect inflammatory-driven neuroaxonal damage and could be used to predict disease activity over the next few years [21]. Today, serum NfL measurement is proposed as a biomarker for several neurological diseases, such as multiple sclerosis, Parkinson’s disease, and Alzheimer’s disease [22]. Recently, the neuro-biomarker Nfl was analyzed in COVID-19 viral disease, and increased serum levels were reported [23,24]. We reported that serum brain-derived neurotrophic factor and NfL and/or their ratios with metalloproteinase-2 and -9 could represent early predictors of neurocovid in COVID-19 patients [24]. Additionally, others have observed that vaccination protects individuals against the estimated disease severity predicted via NfL quantification after COVID-19 hospitalization [25]. The relationship between the mechanisms of neural damage during SARS-CoV-2 infection is not fully understood [26]. Different hypotheses include direct invasion of the central nervous system, endothelial damage, hypercoagulability, systemic inflammation, hypoxia in association with peripheral immune signatures, autoimmunity, and neurodegeneration [27]. Lastly, a new investigation route focuses on the neurological manifestations of long-COVID syndrome [28,29]. All these mechanisms prompted us to explore both the SSHL and serum levels of NfL in our post-COVID-19 patient.

## 2. Case Report

A 67-year-old female presented to the otolaryngology department with a 10-day history of SSHL on the left side without vertigo, dizziness, or tinnitus after a positive test result for SARS-CoV-2. She had received all three doses of the vaccine and she had never tested positive before this episode. She did not report any history of earache, trauma, or discharge and denied any previous hearing loss. She had a medical history of foramen oval closure in 2008 and had been taking cardioaspirin (100 mg/day) ever since. She denied consuming tobacco or alcohol. Otoscopy was insignificant in both ears. Rinne’s test was positive on both sides, while Weber’s test was lateralized to the right ear. Pure-tone audiometry showed left SSHL of at least 50 dB, mostly at high and low frequencies (125, 250, 4000, 6000, 8000 Hz) (Figure 1).

## 3. Investigations, Treatment, and Follow Up

On examination, the chest was clear and the heart sounded normal, and the patient was hemodynamically stable. She developed a fever (37 °C) and dry cough during the first three days after she tested positive. The only other symptom she referred to was sudden hearing loss in the left ear. No other neurological symptoms were reported, and the blood profile was normal. In addition, brain and internal auditory canal magnetic resonance and magnetic resonance angiography were conducted to show any modifications. After the diagnosis of SSHL, before the therapy, we measured the serum NfL levels as 26.1 pg/mL (picogram/milliliter). The patient was managed with intratympanic steroid injections of Dexamethasone 4 mg once a day over three alternating days, repeated for two cycles. Overall, she underwent six intratympanic steroid injections. After completing the treatment, we performed a new pure-tone audiometry test that did not reveal any notable improvement in hearing (Figure 1), and surprisingly, the comparison of the two audiometry tests did not reveal any significant difference. In addition, a new serum measurement of the NfL levels showed an increase in this neuro-biomarker of 31.8 pg/mL (Figure 2). Both serum NfL measurements were higher with respect to the control value (13.3 pg/mL) obtained from *n* = 19 age-matched healthy subjects without post-COVID-19 symptomatology or other neurological diseases associated with NfL increase (Figure 2). Moreover, the patient complained of the onset of slight tinnitus in the left ear. Lastly, after completing the treatment, a third pure-tone audiometry test was executed, revealing no improvement in hearing (data not shown) or differences with the previous audiometry results (Figure 3).

## 4. Discussion

The clinical spectrum of COVID-19 ranges from no symptoms to septic shock and multi-organ dysfunction. The virus causes different upper-respiratory-tract-related symptoms such as nasal congestion, sore throat, and smell or taste dysfunction [30]. A systematic literature review regarding the ENT manifestations of COVID-19 was reported [6], but no ear manifestations were documented. In fact, hearing complication due to coronavirus is rarely mentioned in the literature [31]. An updated meta-analysis demonstrated that hearing loss, tinnitus, and dizziness were statistically significant in patients with COVID-19 [32]. Instead, a previous report on other coronavirus infections documented brainstem involvement and the possibility of neuro-auditory problems [33]. Mustafa et al. showed that COVID-19 could have negative deleterious effects on cochlear hair cell functions despite patients being asymptomatic, as reductions in pure-tone thresholds, as well as transient ear otoacoustic emission amplitudes, were detected [34]. In any case, according to the literature, the non-ENT symptoms are the most common. Meng and colleagues published a systematic review of COVID-19 and SSHL. They reported that growing evidence suggests that COVID-19 carries a risk of developing SSHL, even though its pathogenesis remains unclear [7]. They treated patients with post-COVID-19 SSHL that developed between a few days and 2 months after diagnosis, and glucocorticoid intratympanic administration was considered to reduce its effects.

The majority of COVID-19-related SSHL cases presented showed partial improvement following corticosteroid treatment, but residual deafness was registered in post-COVID-19 patients [35,36]. Our hospital protocol prefers intratympanic administration, eliminating systemic side effects, as compared to intravenous corticosteroids [8,37]. In cases of suspected hearing loss, in the presence of SSHL, hearing tests are suggested, and prompt and aggressive treatment is vital. The nose, nasopharynx, and oropharynges are the main harbor sites of the infection and main sources of its transmission [6]. Furthermore, the virus can invade the central and peripheral nervous systems and can cause different neurological diseases [6]. The annual incidence of SSHL varies between 5 and 30 cases per 100,000 persons in the general population [33]. Its possible causes include viral infection, circulatory deficits, autoimmune disease, labyrinthine membrane rupture, tumors, and central nervous system anomalies. A variety of evidence suggests that viral infection is causative of SSHL. The suggested mechanisms are direct viral access to the labyrinth or the cochlear nerve, reactivation of the latent virus inside the spiral ganglia, and the immunoregulation of systemic viral infections [5]. The virus can induce an immunologic and inflammatory response resulting in direct damage to the cochlea. The neuronal damage and the inflammatory state induced by the virus are therefore reflected both in the clinic (the pure-tone audiogram) and in the altered serum levels of NfL. In this regard, this is the first report that describes an increase in neurofilament light expression associated with SSHL as a potential post-COVID-19 effect or as auditory impairment alone. This is particularly relevant because the level of serum NfL, before (26.1 pg/mL) and after (31.8 pg/mL) the corticosteroid treatment, doubled with respect to the control value (13.3 pg/mL), combined with altered audiometry registration. This NfL increase might be attributed to a damaged auditory nerve, evaluable only via biochemical analysis with a neuro-biomarker of nerve damage (Figure 2). In addition, the anamnestic data do not evidence another concomitant type of post-COVID infection or drug treatment or that the effects of hearing loss were due to a secondary infection with another pathogen.

Potentially, direct virus invasion, an inflammatory response, a microglia reaction, an autophagic mechanism, or another molecular toxic effect could be attributed as a pathogenic stimulus.

Moreover, it is important to note that the cochlear arterial system is terminal, showing several vascularization arborization variants and a microvascular disorder associated with virus entry or inflammation/immunity processes that could cause sudden unilateral hearing loss through an indirect mechanism. The microvascular injury was revealed in the brains of COVID-19 patients through post-mortem histopathological examination, showing degeneration of the endothelial cells’ basal lamina and the congestion of blood vessels, combined with perivascular inflammatory infiltration [33]. In the post-COVID-19 era, many individuals show atypical symptoms, and presumably, any symptoms should be considered in association with previous viral infections. The treatment of post-COVID-19 SSHL and idiopathic SSHL is the same. The administration of corticosteroids within two weeks as initial therapy for SSHL is considered good clinical practice according to the existing guidelines [38].

Furthermore, the intratympanic injection of steroids does not reduce the activation of the immunological response against the virus, and a therapeutic protocol based on the intratympanic injection of steroids, specifically, Dexamethasone 4 mg once a day for three alternate days and repeated for two cycles, may be appropriate for initial SSHL treatment. In the examined case, despite having applied a therapeutic protocol that normally yields excellent results, we did not obtain an improvement in hearing function. This finding is also supported by the evidence showing that serum NfL levels increased over time instead of decreasing, as reported for neurological acute diseases. This could be due the fact that the neuronal damage perpetrated by the virus, despite timely diagnosis and treatment, is irreversible. In accordance with the patient’s wishes, we will perform a new pure-tone audiometry test and a new assessment of the serum NfL levels to evaluate the trend of the patient’s hearing over time.

## 5. Conclusions

The incidence of SSHL is reported to be between 5 and 30 cases per 100,000 [38]. Although SSHL can occur after COVID-19 infection, the etiologic mechanism is still unclear [39]. As a future perspective, we suggest exploring the modifications of potential biomarkers associated with SSHL that can contribute to elucidating the potential biological mechanisms bridging sensory loss and COVID-19, after virus infection and after vaccine administration, aiming to explore a possible overlapping mechanism used by an immune response and consequent nervous damage. The observation of NfL increase associated with hearing loss offers a new perspective on the possibility of monitoring the biochemical profile of this pathological condition. We believe that this could be an interesting field of research and development [2]. Recent studies have described the search for valuable biomarkers in SSHL diagnosis and prognosis, such as HSP70, Prestin, IL-10, serum α-1-acid glycoprotein, galectin-3, and the C-reactive protein to serum albumin ratio [40,41,42], and an association with NfL could be investigated.

Another important issue in explaining this observation could be the evaluation of potential vestibular pathway impairment through vestibular-evoked myogenic potentials (VEMPs) in patients with the recurrent presence of vertigo [43]. This case report demonstrates a temporal association between COVID-19 infection, sudden sensorineural hearing loss, and increased neurofilament expression in post-COVID-19 conditions [44], suggesting ideas for investigations useful for the prognosis of SSHL and supporting potential therapy [1]. This association should be carefully evaluated and not be dismissed as a coincidental event.

## Figures and Tables

**Figure 1 diseases-11-00092-f001:**
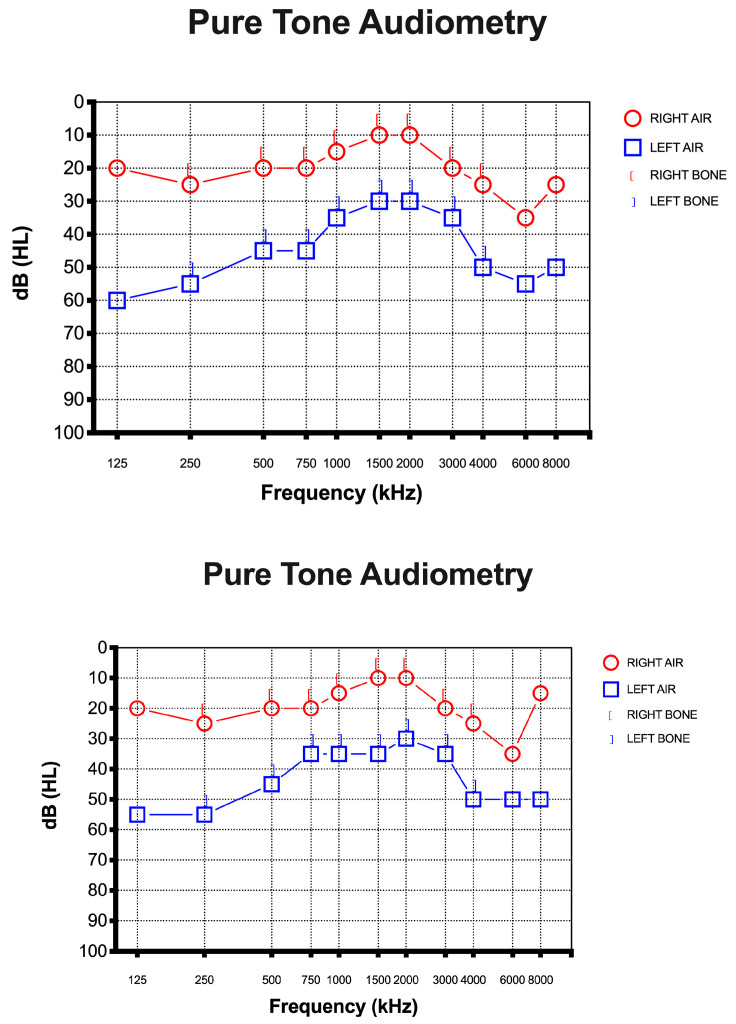
Pure-tone audiometry was performed at the time of diagnosis. The x-axis represents the frequency of sound in hertz (kHz). The y-axis represents the (inverted) intensity of sound in decibels. Hearing level is shown as (HL). (**Up**) Pure-tone audiometry was performed before intratympanic steroid injection. (**Down**) Pure-tone audiometry was performed after intratympanic steroid injection.

**Figure 2 diseases-11-00092-f002:**
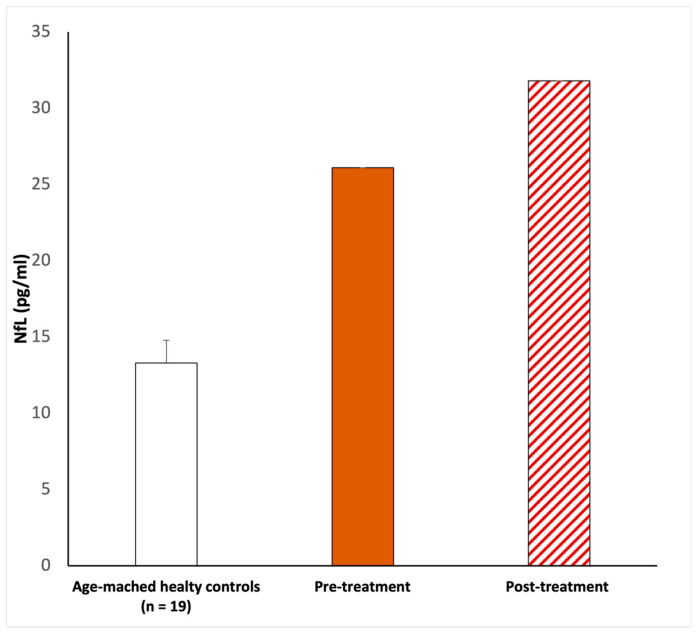
NfL serum level evaluations. The first NfL measurement (26.1 pg/mL) was taken upon pre-treatment. After corticosteroid infiltration treatment, 20 days after, the serum NfL was newly measured (31.8 pg/mL). In both cases, Nfl was elevated with respect to the age-matched healthy controls (13.39 ± 1.5 pg/mL). According to the methods previously described, a peripheral blood sample of 5 mL was collected in a BD Vacutainer™ serum separation tube and centrifuged at 3000 rpm for 15 min. The evaluation of NfL was performed using the Simple PlexTM Ella assay with an EllaTM microfluidic system [23,24], calibrated using the in-cartridge factory standard curve. The limit of detection of NfL was 1.09 pg/mL, calculated by adding three standard deviations to the mean background signal determined from multiple runs.

**Figure 3 diseases-11-00092-f003:**
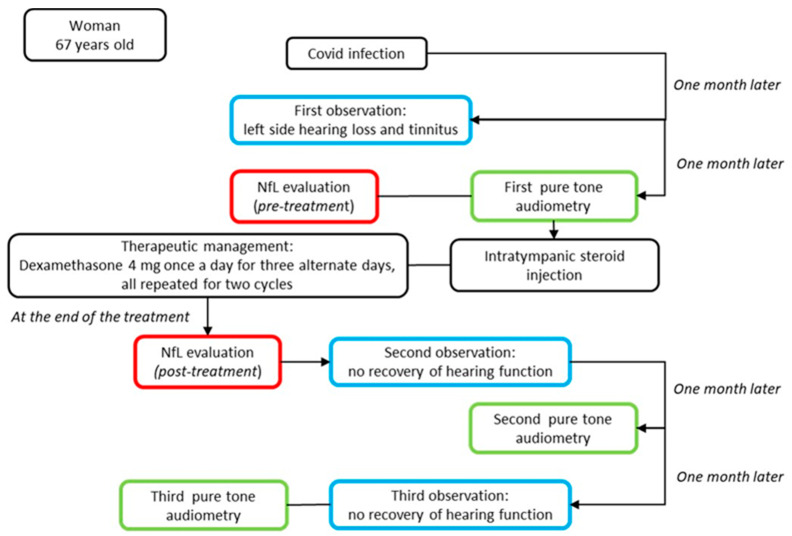
Top-down timeline of the applied clinical–instrumental evaluations and NfL measurements in the SSHL case report. The blue box corresponds to the clinical evaluation; the green box to instrumental exams; and the red box to the serum neurofilament light measurement.

## Data Availability

Data are available upon request.

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
