# Peer review of "Neurofilament Expression as a Biomarker of Post-COVID-19 Sudden Sensorineural Hearing Loss"

_diseases, 2023, doi:10.3390/diseases11030092_

Round 1
Reviewer 1 Report
The serum NfL level of SSNL is only measured in this one case, the reason of the elevated level is unclear.
In the abstract, SSHL is affects is not appropriate, you must omit is in this sentence. There are some points you have to improve English in the manuscript.
Author Response
Dear Editor,
We thank the editor for his/her letter and the reviewers for their comments on our manuscript (Manuscript ID: diseases-2462652). Those comments are all valuable and very helpful for revising and improving our paper, as well as the important guiding significance to our research. We have studied the comments carefully and have made corrections which we hope meet with approval. Revised portions are marked in yellow on the paper. Please see below for point-by-point responses. The main corrections in the manuscript and the response to the reviewer's comments are as follows:
Reply to the Reviewer # 1 comments:
REV1
R1a The serum NfL level of SSNL is only measured in this one case, the reason of the elevated level is unclear.
R1a This is exactly the novelty of this communication. This is the first time that was described an association between SSHL and NfL increased. Obviously, the reason is unknown and it is under investigation. The subject showed only SSHL post-Covid19 infections without any symptoms, signs, or instrumental/ parameters altered, as reported in the manuscript. It is reasonable to suppose a concomitant observation of SSHL and high serum levels of NfL, but the hypnotizable cause of SSHL, (SARS-COV-2), suggests a putative involvement of neuronal structures, as reliable by NfL.
Comments on the Quality of English Language
R1b In the abstract, SSHL is affects is not appropriate, you must omit is in this sentence. There are some points you have to improve English in the manuscript.
R1b The sentence was corrected. ‘is’ was refused. The English language was checked by a professional team.
Thanks again for the reviewer's comments, we revised each quoted sentence again.
We appreciate for Editors/Reviewers’ warm work earnestly and really hope that our modification of this paper can get your precious recognition, which is of great significance to us.
Reviewer 2 Report
The topic is interesting and important. However, there are several key areas that need more work prior to publication. I have summarized the required changes in the hope that the feedback will be useful to you as you update the paper:
1. Line 163: the number of patients in tonsil enlargement is 23. The rest of the listed cases do not contain such information. The content of the given facts should be clarified.
2. In our opinion, the sample used (one case!), on the basis of which the study was conducted and conclusions were drawn, remains incomprehensible.
3. Conclusions should not contain references to primary sources. These are personal conclusions on the basis of the results of the author's research.
Then the conclusions contain information that should be covered in the Introduction (eg, Wichova et al. argue, Recent studies).
Author Response
Dear Editor,
We thank the editor for his/her letter and the reviewers for their comments on our manuscript (Manuscript ID: diseases-2462652). Those comments are all valuable and very helpful for revising and improving our paper, as well as the important guiding significance to our research. We have studied the comments carefully and have made corrections which we hope meet with approval. Revised portions are marked in yellow on the paper. Please see below for point-by-point responses. The main corrections in the manuscript and the response to the reviewer's comments are as follows:
Reply to the Reviewer # 2 comments:
REV2
The topic is interesting and important. However, there are several key areas that need more work prior to publication. I have summarized the required changes in the hope that the feedback will be useful to you as you update the paper:
R1.Line 163: the number of patients in tonsil enlargement is 23. The rest of the listed cases do not contain such information. The content of the given facts should be clarified.
R1. Line 163: number of patients was eliminated, because this data was not reported for other % distribution of ENT manifestations. (The data was reported in [ref. 6] results section and table1 and 2).
R2. In our opinion, the sample used (one case!), on the basis of which the study was conducted and conclusions were drawn,remains incomprehensible.
R2. We thank the reviewer for this observation. The manuscript is a case report, not a pilot study. We are analyzing other SSHL post-covid-19 subjects, but this is the argument of another communication. We retain that the concomitant elevation of Nfl in the serum of SSHL post-Covid-19 patients could be interesting information for the research community stimulating the exploration of new SSHL biomarkers.
R3. Conclusions should not contain references to primary sources. These are personal conclusions on the basis of the results of the author's research.
Then the conclusions contain information that should be covered in the Introduction (eg, Wichova et al. argue, Recent studies).
R3. We thank the reviewer and agree with this suggestion. Line 241: The sentence was deleted and added to the introduction, lines 57-58.
Thanks again for the reviewer's comments, we revised each quoted sentence again.
We appreciate for Editors/Reviewers’ warm work earnestly and really hope that our modification of this paper can get your precious recognition, which is of great significance to us.
Reviewer 3 Report
This manuscript described the relationship among the sudden sensorineural hearing loss (SSHL), the serum levels of the neurofilament light chain (NfL) protein, the treatment of intratympanic steroid administration and with the post-Covid-19 infection. This is the otorhinolaryngology paper.
Even the COVID-19 pandemic has been gradually restored, the more medical therapeutic information should be corrected from various perspectives and areas. In this regard, this may be a one of valuable information. However, several problematic points were found in a current state.
1. Anyway, unfortunately, I felt strongly that English related errors are frequently observed and these missing disrupted a correct reading of this manuscript and/or it makes it illegible.
2. Basically, many abbreviations were used throughout the text, so authors would have to take care of them.
3. What is the point of figure 1 and 2? This means that any significant improvement in hearing has been observed following Dexamethasone treatment. In that case, it is not necessary to have individual presentations. Outlined in the text is sufficient, and it can be deleted Figure 2 or include in Figure 1.
4. The enumerator descriptions are noted (such as tautology and or thesaurus). How does the use of flowchart (like Figure 4) to the expert diagnosis and healing information? This may be helpful for the reader and be an additional function (total number of Figures is not changing according to the above 3).
5. For Figure 3; The serum NfL showed the trend in increase following aging. Thus, the values of normal age control measured by the same analysis (technique) should be added. In this regard, “Healthy Control” should change to “Age matched healthy Control”, to avoid (reduced) miss understanding.
6. (L 33) Spell out SARS-CoV-2
7. (L 42) Insert ENT into the parenthesis = ear nose and throat (ENT)
8. What is a difference SSHL and SSNHL? Simply miss typing? If not, should be spell out.
9. (L 133) 31,8 pg/ml what is it? Numerical representation is basically wrong.
Similarly, the miss usage of comma and periods are frequently seen throughout the text.
1. Anyway, unfortunately, I felt strongly that English related errors are frequently observed and these missing disrupted a correct reading of this manuscript and/or it makes it illegible.
2. Basically, many abbreviations were used throughout the text, so authors would have to take care of them.
3. (L 33) Spell out SARS-CoV-2
4. (L 42) Insert ENT into the parenthesis = ear nose and throat (ENT)
5. What is a difference SSHL and SSNHL? Simply miss typing? If not, should be spell out.
6. (L 133) 31,8 pg/ml what is it? Numerical representation is basically wrong.
7. Similarly, the miss usage of comma and periods are frequently seen throughout the text.
Author Response
Dear Editor,
We thank the editor for his/her letter and the reviewers for their comments on our manuscript (Manuscript ID: diseases-2462652). Those comments are all valuable and very helpful for revising and improving our paper, as well as the important guiding significance to our research. We have studied the comments carefully and have made corrections which we hope meet with approval. Revised portions are marked in yellow on the paper. Please see below for point-by-point responses. The main corrections in the manuscript and the response to the reviewer's comments are as follows:
Reply to the Reviewer # 3 comments:
REV3
This manuscript described the relationship among the sudden sensorineural hearing loss (SSHL), the serum levels of the neurofilament light chain (NfL) protein, the treatment of intratympanic steroid administration and with the post-Covid-19 infection. This is the otorhinolaryngology paper.
Even the COVID-19 pandemic has been gradually restored, the more medical therapeutic information should be corrected from various perspectives and areas. In this regard, this may be a one of valuable information. However, several problematic points were found in a current state.
R1. Anyway, unfortunately, I felt strongly that English related errors are frequently observed and these missing disrupted a correct reading of this manuscript and/or it makes it illegible.
R1. The Quality of the English Language was improved and many typos and grammatical errors were corrected. The manuscript was revised by a native English colleague.
R2. Basically, many abbreviations were used throughout the text, so authors would have to take care of them.
R2. The following abbreviations were replaced: Ear, nose, and throat (ENT); brain derived-neurotrophic factor (BDNF) metalloproteinase- 2 and -9 MMP-2 and MMP-9; central nervous system (CNS), magnetic resonance imaging (MRI); magnetic resonance angiography (MRA); α‑1‑acid glycoprotein (AGP),
R3. What is the point of figure 1 and 2? This means that any significant improvement in hearing has been observed following Dexamethasone treatment. In that case, it is not necessary to have individual presentations. Outlined in the text is sufficient, and it can be deleted Figure 2 or include in Figure 1.
R3. Thank you very much for this observation. As suggested, we deleted Figure 2 and include it in Figure 1.
R4. The enumerator descriptions are noted (such as tautology and or thesaurus). How does the use of flowchart (like Figure 4) to the expert diagnosis and healing information? This may be helpful for the reader and be an additional function (total number of Figures is not changing according to the above 3).
R4. The term flowchart was eliminated from Fig. 3 and the footnote description was rewritten. The Fig. 3 timeline supports the text description, and we are not able to suggest a flowchart of the expert diagnosis and healing information. We are collecting several other cases, and at the end of the study, many questions will be cleared.
R5. For Figure 3; The serum NfL showed the trend in increase following aging. Thus, the values of normal age control measured by the same analysis (technique) should be added. In this regard, “Healthy Control” should change to “Age matched healthy Control”, to avoid (reduced) miss understanding.
R5. Line 147: the sentence (‘Age-matched healthy Control’) is already present in the original text.
R6. (L 33) Spell out SARS-CoV-2
R6. Line 33: Severe Acute Respiratory Syndrome COronaVirus 2 (SARS-CoV-2)
R7. (L 42) Insert ENT into the parenthesis = ear nose and throat (ENT)
R7. Line 42: we inserted ENT into the parenthesis.
R8. What is a difference SSHL and SSNHL? Simply miss typing? If not, should be spell out.
R8. Thank you for the miss typing of SSHNL.
R9. L 133) 31,8 pg/ml what is it? Numerical representation is basically wrong.
R9. The pg/ml correspond to (picogram/milliliter) as reported in line 132. The quantification of the Nfl by methods reported in the text were numerically indicated as picogram/milliliter by Simple PlexTM Ella assay (ProteinSimple, San Jose, CA, USA) on EllaTM microfluidic system (Bio-Techne, Minneapolis, MN, USA).
Similarly, the miss usage of comma and periods are frequently seen throughout the text.
An extensive evaluation of comma periods was done throughout the text.
Thanks again for the reviewer's comments, we revised each quoted sentence again.
We appreciate for Editors/Reviewers’ warm work earnestly and really hope that our modification of this paper can get your precious recognition, which is of great significance to us.
Reviewer 4 Report
The authors wrote an interesting case report about Evaluation of Neurofilament as Biomarker of Post-COVID-19 Sudden Sensorineural Hearing Loss. The case is well written, very interesting and the topic is hot. Surely is worthy of publication.
I have some suggestion to increase the scientific impact of the manuscript and to give a better evidence in literature.
1. The nerve injury is surely a possible cause of SSHL, so I think that is also important evalue the vestibular nerve thanks to VEMPs. Please use this reference in the discussion: Gazia F, Galletti B, Freni F, Bruno R, Sireci F, Galletti C, Meduri A, Galletti F. Use of intralesional cidofovir in the recurrent respiratory papillomatosis: a review of the literature. Eur Rev Med Pharmacol Sci. 2020 Jan;24(2):956-962.
2. Please in the conclusion stressed the fact that the neurofilament biomarker is useful for the PROGNOSIS of SSHL, not for valuing the potential therapy. Please correct this sentence in conclusion and other parts of the manuscript.
Author Response
Dear Editor,
We thank the editor for his/her letter and the reviewers for their comments on our manuscript (Manuscript ID: diseases-2462652). Those comments are all valuable and very helpful for revising and improving our paper, as well as the important guiding significance to our research. We have studied the comments carefully and have made corrections which we hope meet with approval. Revised portions are marked in yellow on the paper. Please see below for point-by-point responses. The main corrections in the manuscript and the response to the reviewer's comments are as follows:
Reply to the Reviewer # 4 comments:
We thank the editor for his/her letter and the reviewers for their comments on our manuscript (Manuscript ID: diseases-2462652).
REV4
The authors wrote an interesting case report about Evaluation of Neurofilament as Biomarker of Post-COVID-19 Sudden Sensorineural Hearing Loss. The case is well written, very interesting and the topic is hot. Surely is worthy of publication.
I have some suggestion to increase the scientific impact of the manuscript and to give a better evidence in literature.
R4a The nerve injury is surely a possible cause of SSHL, so I think that is also important evalue the vestibular nerve thanks to VEMPs. Please use this reference in the discussion: Gazia F, Galletti B, Freni F, Bruno R, Sireci F, Galletti C, Meduri A, Galletti F. Use of intralesional cidofovir in the recurrent respiratory papillomatosis: a review of the literature. Eur Rev Med Pharmacol Sci. 2020 Jan;24(2):956-962. doi: 10.26355/eurrev_202001_20081.
R4a. We thank the reviewer for this suggestion and the vestibular nerve evaluation by VEMPs was added in the text. The reference suggested was added as [45].
R4b. Please in the conclusion stressed the fact that the neurofilament biomarker is useful for the PROGNOSIS of SSHL, not for valuing the potential therapy. Please correct this sentence in the conclusion and other parts of the manuscript.
R4b. In the conclusion, the sentence was replaced with: “suggesting investigations useful for the prognosis of SSHL, supporting potential therapy [47].”. In addition, we will evaluate after 1 year from the SSHL diagnosis, all parameters described in this case report, and as suggested, also the VEMPs.
Thanks again for the reviewer's comments, we revised each quoted sentence again.
Round 2
Reviewer 2 Report
The authors have made a significant effort and addressed all issues from the previous review round, thus improving the quality of their paper. I suggest the acceptance of the paper in its present form.
Author Response
REVIEWER 2-II ROUND
Comment The authors have made a significant effort and addressed all issues from the previous review round, thus improving the quality of their paper. I suggest the acceptance of the paper in its present form.
Response Thanks again for the reviewer's comments. We appreciate for Reviewers’ warm work earnestly and really hope that our modification of this paper can get your precious recognition, which is of great significance to us.
Reviewer 3 Report
After revision, the manuscript improved somewhat, but the authors still did not respond to my important suggestion, which regarded it as major revisions. Thus, responses were unsatisfactory.
First, the authors fail to understood the addition of the “normal healthy control values” (probably 20-40 years). This is different from “age matched healthy control” (they were older people around 67-70 years). This is important to clarify the age dependent increase in NFL. Refer to following.
Comment 5. For Figure 3; The serum NfL showed the trend in increase following aging. Thus, the values of normal age control measured by the same analysis (technique) should be added. In this regard, “Healthy Control” should change to “Age matched healthy Control”, to avoid (reduced) miss understanding.
Response 5. Line 147: the sentence (‘Age-matched healthy Control’) is already present in the original text.
For the above; The authors misunderstood. My point is adding the normal age control value in this graph, because the serum NfL showed the trend in increase following aging. In this regard, Healthy control should change to Age matched control in the graph (not in the legend). The normal control means young healthy adult, probably 20-40 years-old using the same analysis (technique) of current study. The number of (N=19 of control) should be also indicated in the figure. Similarly, why the values in the figure legend did not have SD should present. I certainly want to see the above "age depend differences".
Second, I am hard to believe that a careful English editing was done. There were some very, very fundamental errors left. It seems to me that the author takes my remarks lightly. Authors should know that take seriously the attitude toward international publishing in English.
Comment 9. L 133) 31,8 pg/ml what is it? Numerical representation is basically wrong.
Response 9. The pg/ml correspond to (picogram/milliliter) as reported in line 132. The quantification of the Nfl by methods reported in the text were numerically indicated as picogram/milliliter by Simple PlexTM Ella assay (ProteinSimple, San Jose, CA, USA) on EllaTM microfluidic system (Bio-Techne, Minneapolis, MN, USA).
Comment 10. Similarly, the miss usage of comma and periods are frequently seen throughout the text.
Response 10. An extensive evaluation of comma periods was done throughout the text.
For the above; there is missing the whole point. I mean 31,8 is wrong, 31.8 is correct. I have never seen such mistake. There are only triple digits. Professional English editor never missing such problem.
Thus, I can’t accept the Response 1 such as; “The Quality of the English Language was improved and many typos and grammatical errors were corrected. The manuscript was revised by a native English colleague”. I can’t believe that the native undetected such mistakes.
(L 160-161) present in 2.1% of patients, nasal congestion in 4.1%, smell affection in patients 6%, nasal obstruction in 3.4%, sore throat in 11.3%, pharyngeal erythema 5.3%
(L 197-199) This is particularly relevant because the serum NfL before (26,1 pg/ml) and after (31,8 pg/ml) the corticoster-198 treatment, was doubled with respect to the control value (13,3 pg/ml). Similarly, in the legend of Figure 2 include the same mistake except for 1.09 pg/ml. Consist (ml) not (mL).
Author Response
Dear Editor,
We thank the editor for his/her letter and the reviewers for their comments on our manuscript (Manuscript ID: diseases-2462652). Those comments are all valuable and very helpful for revising and improving our paper, as well as the important guiding significance to our research. We have studied the comments carefully and have made corrections which we hope meet with approval. Revised portions are marked in yellow on the paper. Please see below for point-by-point responses. The main corrections in the manuscript and the response to the reviewer's comments are as following:
REVIEWER 3-II ROUND
After revision, the manuscript improved somewhat, but the authors still did not respond to my important suggestion, which regarded it as major revisions. Thus, responses were unsatisfactory.
First, the authors fail to understood the addition of the “normal healthy control values” (probably 20-40 years). This is different from “age matched healthy control” (they were older people around 67-70 years). This is important to clarify the age dependent increase in NFL. Refer to following.
Comment 5. For Figure 3; The serum NfL showed the trend in increase following aging. Thus, the values of normal age control measured by the same analysis (technique) should be added. In this regard, “Healthy Control” should change to “Age matched healthy Control”, to avoid (reduced) miss understanding.
Response 5. Line 147: the sentence (‘Age-matched healthy Control’) is already present in the original text.
Comment IIR. For the above; The authors misunderstood. My point is adding the normal age control value in this graph, because the serum NfL showed the trend in increase following aging. In this regard, Healthy control should change to Age-matched control in the graph (not in the legend). The normal control means young healthy adult, probably 20-40 years-old using the same analysis (technique) of current study. The number of (N=19 of control) should be also indicated in the figure. Similarly, why the values in the figure legend did not have SD should present. I certainly want to see the above "age depend differences".
Response IIR. Thanks again for the reviewer's comments and the clarification of the comment. We apologize for the misunderstanding. We changed “Healthy Control” to “Age-matched healthy Control”, in the new figure 2. The number of (N=19 of control) was indicated in the figure, as the values of SD.
Comments on the Quality of English Language
Comment, Second, I am hard to believe that a careful English editing was done. There were some very, very fundamental errors left. It seems to me that the author takes my remarks lightly. Authors should know that take seriously the attitude toward international publishing in English.
Response. We are very sorry for the low quality of the English language in the manuscript.
We also share the doubt that the manuscript has been reviewed by a native English-speaking colleague, and we will make the appropriate checks.
We apologize for this inconvenience and will request, as suggested by the journal, that English-language editing of the final version of the manuscript can be arranged by the MDPI service.
Comment 9. L 133) 31,8 pg/ml what is it? Numerical representation is basically wrong.
Response 9. The pg/ml corresponds to (picogram/milliliter) as reported in line 132. The quantification of the Nfl by methods reported in the text was numerically indicated as picogram/milliliter by Simple PlexTM Ella assay (ProteinSimple, San Jose, CA, USA) on EllaTM microfluidic system (Bio-Techne, Minneapolis, MN, USA).
Comment 10. Similarly, the miss usage of comma and periods are frequently seen throughout the text.
Response 10. An extensive evaluation of comma periods was done throughout the text.
Comment For the above; there is missing the whole point. I mean 31,8 is wrong, 31.8 is correct. I have never seen such mistake. There are only triple digits. Professional English editor never missing such problem.
Response. We are sorry for these errors. The commas were replaced by a point as suggested.
Comment Thus, I can’t accept the Response 1 such as; “The Quality of the English Language was improved and many typos and grammatical errors were corrected. The manuscript was revised by a native English colleague”. I can’t believe that the native undetected such mistakes.
(L 160-161) present in 2.1% of patients, nasal congestion in 4.1%, smell affection in patients 6%, nasal obstruction in 3.4%, sore throat in 11.3%, pharyngeal erythema 5.3%
Response. The sentence (L 160-161) was replaced with: “A systematic literature review regarding ENT manifestations for COVID-19 was reported [6], but no ear manifestations were documented. In fact, hearing complications due to coronavirus is little mentioned in the literature [31].”
Comment (L 197-199) This is particularly relevant because the serum NfL before (26,1 pg/ml) and after (31,8 pg/ml) the corticosteroid treatment, was doubled with respect to the control value (13,3 pg/ml). Similarly, in the legend of Figure 2 include the same mistake except for 1.09 pg/ml. Consist (ml) not (mL).
Response. We are sorry for these errors. The comma was replaced by a point as suggested, and (mL) was corrected (ml).
We appreciate for Editors/Reviewers’ warm work earnestly and really hope that our modification of this paper can get your precious recognition, which is of great significance to us.
Round 3
Reviewer 3 Report
No comments